# Elucidating the Mesocarp Drupe Transcriptome of Açai (*Euterpe oleracea* Mart.): An Amazonian Tree Palm Producer of Bioactive Compounds

**DOI:** 10.3390/ijms24119315

**Published:** 2023-05-26

**Authors:** Elaine Darnet, Bruno Teixeira, Hubert Schaller, Hervé Rogez, Sylvain Darnet

**Affiliations:** 1Centre for Valorization of Amazonian Bioactive Compounds (CVACBA) & Institute of Biological Sciences, Federal University of Pará (UFPA), Belém 66075-750, PA, Brazil; elaine.darnet@outlook.com (E.D.); bruno.teixeira@icen.ufpa.br (B.T.); sylvain.darnet@univ-lorraine.fr (S.D.); 2International Associated Laboratory PALMHEAT, Frech Scientific Research National Center (CNRS)/UFPA, 75016 Paris, France; hubert.schaller@ibmp-cnrs.unistra.fr; 3Plant Isoprenoid Biology, Institute of Molecular Biology of Plants of the Scientific Research National Center, Strasbourg University, 67081 Strasbourg, France

**Keywords:** anthocyanins, tocopherols, fruit ripening, antioxidant, arecaceae

## Abstract

*Euterpe oleracea* palm, endemic to the Amazon region, is well known for açai, a fruit violet beverage with nutritional and medicinal properties. During *E. oleracea* fruit ripening, anthocyanin accumulation is not related to sugar production, contrarily to grape and blueberry. Ripened fruits have a high content of anthocyanins, isoprenoids, fibers, and proteins, and are poor in sugars. *E. oleracea* is proposed as a new genetic model for metabolism partitioning in the fruit. Approximately 255 million single-end-oriented reads were generated on an Ion Proton NGS platform combining fruit cDNA libraries at four ripening stages. The de novo transcriptome assembly was tested using six assemblers and 46 different combinations of parameters, a pre-processing and a post-processing step. The multiple k-mer approach with TransABySS as an assembler and Evidential Gene as a post-processer have shown the best results, with an N50 of 959 bp, a read coverage mean of 70x, a BUSCO complete sequence recovery of 36% and an RBMT of 61%. The fruit transcriptome dataset included 22,486 transcripts representing 18 Mbp, of which a proportion of 87% had significant homology with other plant sequences. Approximately 904 new EST-SSRs were described, and were common and transferable to *Phoenix dactylifera* and *Elaeis guineensis*, two other palm trees. The global GO classification of transcripts showed similar categories to that in *P. dactylifera* and *E. guineensis* fruit transcriptomes. For an accurate annotation and functional description of metabolism genes, a bioinformatic pipeline was developed to precisely identify orthologs, such as one-to-one orthologs between species, and to infer multigenic family evolution. The phylogenetic inference confirmed an occurrence of duplication events in the Arecaceae lineage and the presence of orphan genes in *E. oleracea*. Anthocyanin and tocopherol pathways were annotated entirely. Interestingly, the anthocyanin pathway showed a high number of paralogs, similar to in grape, whereas the tocopherol pathway exhibited a low and conserved gene number and the prediction of several splicing forms. The release of this exhaustively annotated molecular dataset of *E. oleracea* constitutes a valuable tool for further studies in metabolism partitioning and opens new great perspectives to study fruit physiology with açai as a model.

## 1. Introduction

Palms are one of the emblems of Amazonia. Some of these trees are known worldwide such as the members of the Euterpe genus. These plants were for many years the principal source of palm hearts for human consumption [1]. The açai beverage, a watery emulsion prepared with the fleshy purple pulp of the fruit, is traditionally consumed by indigenous and rural inhabitants of South America. This product is now widely accepted on the Brazilian domestic and international market [2]. Brazilian statistics report an impressive acai production area of approximately 208,111 hectares in 2021 [3]. During the same period, Brazilian exports of açai pulp reached a total of 1.4 million tons, corresponding to USD 1.07 billion, which is approximately twice the production value recorded in 2018 (USD 0.55 billion). *Euterpe oleracea* Martius is the most commercialized species [2] (Appendix A). Its pulp is oleaginous, 50% of its dry matter (DM) is made of lipids, and it has a high content of dietary fibers (21.8% of DM), proteins (10% of DM) and some minerals [4]. The most valuable feature of the pulp is its rich and diverse content of bioactive compounds such as phenylpropanoids and isoprenoids. Therefore, açai was coined “superfruit” by some authors [2,5]. The antioxidant, anti-inflammatory, antileishmania and anti-cancer properties of *E. oleracea* have opened new markets for this product as a natural and healthy diet supplement. For the same reasons, this plant also offers perspectives for further research in pharmacology and biotechnology [5,6,7].

Despite the nutritional, economical and biotechnological importance of *E. oleracea*, little is known about the genetic. At the molecular genetics level, 133 marker sequences obtained in taxonomy studies are found in the National Center for Biotechnology Information database (NCBI) (accessed on 14 April 2023). The phylogenetic relationship of *E. oleracea* with other palms whose genomes were sequenced is shown in Figure 1
*Elaeis guineensis* (African oil palm) in Cocoseae and *Phoenix dactylifera* (date palm) are included in Phoeniceae tribes. *E. oleracea* has diverged from the Euterpeae tribe 100 Ma ago [8]. Recently, a cytogenetic characterization of the Euterpe genus revealed that three species, i.e., *E. oleracea*, *E. edulis* and *E. precatoria*, were commonly confounded due to their highly similar morphology [9]. These species exhibit a distinct geographical distribution in Eastern, Western Amazon, and Atlantic forest, respectively. Additionally, they possess the same chromosome number (2n = 36), but their chromosome morphology differs [9]. In the same study, the genome size of *E. oleracea* was estimated at 4.22 pg in C-value, corresponding to 4.13 Gbp [9]. This diploid genome is larger than the human genome, one of the largest eukaryotic genomes known so far. Repetitive sequences found in palm trees are covering more than 40% of the genome, as reported for *E. guineensis* [10]. A thorough characterization of the genome of *E. oleracea* fruit is of tremendous importance for several reasons. Genomic data and associated knowledge will grant access to biological diversity and gene discovery in line with the high agronomical value of the fruit and consequently will benefit the agricultural and biotechnological outcomes. Next-generation high-throughput RNA sequencing (RNA-Seq) is a powerful and fast tool for gene discovery in transcriptome datasets of non-model plants [11]. De novo assembly of eukaryotic transcriptomes or genomes using short reads is one of the most significant challenges in bioinformatics [12]. For each newly sequenced species, the de novo assembly method needs to be (re) optimized according to the library preparation protocol, sequencing chemistry, biological and genomic characteristics of the samples [13]. Additionally, two processing steps, pre- and post-assembly, have a significant influence on the completeness of a transcriptome prediction [14]. A wide variety of assemblers has been developed based on different algorithms and with specific parameters for each next-generation sequencing platform (NGS) [15]. When facing such a diversity of bioinformatic tools, the current trend would be first to optimize the parameters for each assembler and in a next step, to compare and merge the outputs to generate a final transcriptome with the highest completeness. These combining approaches are based on hybrid assembly, multi-assembler and multiple *k*-mer additive methods [16,17,18,19,20,21]. The most critical point of de novo assembly is the quality of the transcriptome reconstruction, especially its completeness and the absence of non-biological artifacts or chimeric transcripts [15]. Two kinds of metrics can be used for this matter. Intrinsic metrics are statistics retrieved from transcripts sequences directly, such as transcripts number, length (N50, average), total bases and ORF predictions. Extrinsic metrics are based on read coverage and a comparison of the transcript with the original read dataset or with a proteome or transcriptome from another species. In that case, the comparison is figured out by the Reads Mapping Back to the Transcripts parameter (RMBT). Comparison methods were developed using Basic Local Alignment Search Tool (BLAST) or Reciprocal Blast Hit (RBH) with a unique reference proteome, such as in the Transrate tool [22]. In a given pipeline, this unique reference is substituted by a dataset of core proteins, specific to a taxonomic group, such as in the Benchmarking Universal Single-Copy Orthologs pipeline (BUSCO) [23].

Francisconi et al. (2022) conducted a study with the chloroplast genome sequences of 55 palm species, specifically Euterpe species, identifying polymorphisms and performing a phylogenomic inference. The chloroplast genomes showed a conserved structure with high synteny among other palms. The study also identified SNPs and estimated genetic diversity using nuclear and mitochondrial reads, showing high divergence among species, especially between *E. edulis* and *E. precatoria*. The study concluded that despite few structural differences in the chloroplast genomes, point mutations can differentiate *E. edulis* from other Euterpe species. The resources generated can be used for future phylogenomic inferences and comparative analyses within Arecaceae [24].

Lopes et al. (2021) conducted a study comparing the complete plastomes of two species of Euterpe palm trees, *E. edulis* and *E. oleracea*, to investigate the effects of different environments on evolutionary characteristics. They analyzed various factors such as plastome structure, SSRs, tandem repeats, SNPs, indels, hotspots of nucleotide polymorphism, codon Ka/Ks ratios, and RNA editing sites. The study found 303 SNPs, 91 indels, and 82 polymorphic SSRs between the two species. The study also found a close correlation between the localization of repetitive sequences and indels, suggesting that replication slippage may play a role in plastid DNA mutations in Euterpe. Additionally, the study identified several positive signatures in several plastid proteins in *E. edulis* that were observed as amino acid variants across the palm phylogeny [25].

The functional annotation of a plant transcriptome is a challenge for non-model plants because insufficient genomic resources are available in public databases [26,27]. It is also evident that most of the plants that present ecological and economic value are non-model plants [11,28]. It is, however, crucial to investigate their unique catalog of genes. The identification of gene orthology is a pivotal approach in genomics to infer evolution events, genetic diversity and gene function, mainly in plant species with limited genomic information [29]. The functional prediction by homology is often performed using BLAST or RBH methods. The limitation of these methods is the misprediction of many paralogs as orthologs, especially for a gene that has diverged more rapidly in one of the species [30,31]. Pipelines for orthology assignment are based on tree-based methods, such as PANTHER and Phylome, or on pairwise comparisons, such as EggNOG, Hieranoid, InParanoid, OMA, OrthoInspector. The last cited pipelines are often chosen because of its accurate orthology prediction and its low rate of false positive prediction, compared to other classical ortholog prediction tools [32,33,34]. For studies dealing with plant bioactive products, often called secondary or specialized metabolites, accurate identification of orthologs and paralogs is 118 crucial to understand the evolution of metabolic pathways and the rise of chemodiversity in plant lineages [35]. The OMA pipeline is based on an all-against-all Smith–Waterman alignment method for evaluating the evolutionary distances and inferring orthologous pairs, and a benchmark study shows that this tool has the lower rate of false positive prediction [32,34]. The inferred orthologs are clustered in OMA groups, direct orthologous pairs between species, and Hierarchical Orthologous Group (HOG), grouping genes distributed in a specific taxonomic group and descended from a common ancestor [36]. This new approach for functional annotation should increase our understanding of metabolism pathway evolution, especially in non-model plants.

Due to the large genome size of *E. oleracea*, and a high percentage of repetitive regions in this genome as in other palms, a transcriptomic approach to unveiling a network of genes and their products represents a realistic alternative to sequencing the whole genome [37]. Transcriptome assembly was the next crucial step towards the generation of sound sequence information. *E. oleracea* transcriptome was sequenced to establish a complete catalog of the genes expressed in the fruit, particularly genes related to the biosynthesis of high-value bioactive metabolites [5]. Consequently, a transcriptome profile of *E. oleracea* fruit will support further research in the biology of palms and will help to drive breeding approaches and biotechnology programs. For this, it is required to identify all transcripts related to the production of natural products of interest and to identify molecular markers.

## 2. Results

### 2.1. Library Preparation and Sequencing

NGS of genomes is a powerful tool to access genetic information without prior knowledge and molecular studies, for instance in the case of forest tree species, as *E. oleracea* [38]. Similar to many fruits, *E. oleracea* has a low content of RNAs [39]. Therefore, the extraction of all pulp mass from each fruit was necessary to retrieve more than 10 μg of total 142 RNAs, with RNA Integrity Numbers (RIN) value higher than 7.0 as recommended when using the strand-specific mRNA-Seq sequencing protocol. The cDNA libraries preparation and strand-specific mRNA-Seq sequencing were performed on the Ion Proton platform. In total, 255 millions of single-end reads were obtained. Their size range was from 25 to 300 bp. The total transcriptome sequence size was 17.9 Gbp. Approximately 76% of bases showed a sequencing error rate lower than 1% (QV20). A similar percentage of QV20 was previously observed in datasets generated by the Ion Proton platform [40].

### 2.2. Optimization for De Novo Assembly of E. oleracea Fruit Transcriptome

The bioinformatic analysis is described in details in Figure 2. The raw reads merged into one file are the input for subsequent analysis (Figure 2a). The pre-assembly step (Figure 2b) was performed to eliminate adapter sequences, to trim or remove reads with low sequencing quality, and to remove redundant reads [41]. After this first step, 25 millions of high-quality non-redundant reads were obtained. The quality value was higher than 20 (sequencing error probability of 1%) for 93% of the bases of this 2 Gbp sequence dataset. The optimization of de novo assembly developed in our study is based on a single *k*-mer and multiple *k*-mer additive approaches, using seven different assembler tools. Firstly, several assemblers were tested with different *k*-mers, from *k*-19 to *k*-99 (Figure 2b), except MIRA, Trinity and CLC genomics workbench that require a pre-determined *k*-mer (Figure 2b). Secondly, multiple *k*-mer additive approaches were used to merge transcriptomes obtained by the same assembler with the different *k*-mers (Figure 2b). Finally, 97 transcriptomes were generated (Figure 2b), and 47 metrics were determined to compare and to assess the quality and completeness of each transcriptome. Using *E. guineensis* proteome as a reference, 18 intrinsic and 17 extrinsic metrics were determined with Transrate, 6 extrinsic metrics with BUSCO plant proteome dataset, and 6 intrinsic metrics with bowtie and SAMtools to to evaluate read coverage and RMBT parameters (Appendix A). 

A high variation in metrics between assemblers and *k*-mer approaches was observed. For example, the N50 value ranged from 252 to 959 bp, the total transcriptome size from 167 0.8 Mbp to 130 Mbp, the transcript number from 1732 to 130,423, the percent of BUSCO complete universal single-copy orthologs (CUSCO) from 1 to 36%, and the RMBT value from 22 to 83%. The quality and completeness of transcriptomes were plotted based on N50 value, RMBT, and percent of complete universal single-copy orthologs retrieved from plant BUSCO dataset (CUSCOpB) (Figure 3). N50, RMBT, and CUSCOpB are three parameters widely used as references in recent studies in which they are described as informative indicators to evaluate the quality of transcriptome reconstruction [21,42,43]. Most of the transcriptomes analyzed here have an N50 value below 580 bp and an RBMT value below 70% (Figure 3). These transcriptomes were obtained using unique *k*-mer methods with SOAPdenovo, Velvet/Oases, RNA spades, MIRA and CLCg assemblers. For RNAspades and Velvet/Oases assemblers, the highest values of N50, RMBT and of CUSCOpB were obtained with a *k*-mer of 39 (Figure 3). For SOAPdenovo, a single *k*-mer method of 39 (*k*-mer 39) gave the highest N50 and RMBT. However, *k*-mer 19 showed a slightly higher percentage of CUSCOpB. MIRA has demonstrated a high percentage (approximately 23%) of CUSCOpB, and CLCg a low percentage (approximately 2%). Results obtained with TransABySS were similar to those obtained with the previously described assemblers, with the highest N50 at *k*-mer 43 and the highest CSCOB and RMBT at *k*-mer 29. In this case, the RMBT value was approximately 79% of total reads. The same holds true when the transcriptome assembly was performed by Trinity. In fact, a high RMBT was indicative of high usage of reads. The CUSCOpB was lower with Trinity than with other assemblers. Here, Trinity was developed to efficiently predict all variants of the same transcript, such as splicing variants [44]. In fact, depending on read datasets and the type of sequencing platform, the tendency is to generate an extensive collection of isoforms for the same gene [45]. Statistics about Trinity assembly revealed the prediction of 130,000 transcripts and 19% of duplicated universal single-copy orthologs of plant BUSCO dataset while TransABySS with *k*-mer 23 has only 2% of duplicated orthologs. Altogether our results showed a low N50 value (below 580 bp) compared to other transcriptome studies performed in palm trees. In the case of de novo genome assembly of *Cocos nucifera* and *Nypa fruticans*, N50 values of 1219 bp and 192 1096 bp, respectively, were obtained [46,47]. The improvement of açai fruit transcriptome reconstruction was made using multiple *k*-mer additive approach, combining the single *k*-mer outputs from each different assembler. As described by Surget-Groba and Montoya-Burgos (2010) [20], a bioinformatic analysis that relies on different *k*-mer presents the considerable advantage to detect at the same time low expressed genes and long contigs.

Results obtained with the multiple *k*-mer additive methods provided a transcriptome that is based on a considerable increase in both N50 and CUSCOpB percent value. The best N50 and CUSCOpB values were obtained with TransABySS assembler (Figure 3). This result is most probably due to the high values of CUSCOpB and RBMT obtained for each of the single *k*-mer analysis. With SOAPdenovo, Velvet/Oases and RNAspades, the N50 values were two-fold higher than those obtained with the single *k*-39 method, and the RMBT values were slightly the same in all cases. The CUSCOpB percent has only a real gain for SOAPdenovo and Velvet/Oases. Multiple *k*-mer additive transcriptome based on TransABySS has the higher value of N50, RBMT, and CUSCOB. The decrease in RMBT for multiple *k*-mer additive transcriptomes compared to *k*-mer 29 or *k*-mer 43 transcriptomes could be explained by the transcript assemblies combining step. This step was done with the tr2aacds mRNA Transcript Assembly Software (Evidential Gene pipeline- http://arthropods.eugenes.org/EvidentialGene/trassembly.html, accessed on 18 Mach 2022) that performs a transcript classification based on CDS-DNA local alignment identity, dropping contigs with perfectly identical CDS sequences and maintaining contig with the difference in CDS sequences. The UTR sequences are not considered for assemblies combining step and consequently allows to remove many mis-assemblies contigs but decreases the number of reads mapped back to the assembled transcriptome. The high-quality de novo assembly generated by Trans-ABySS produced 18 Mb of data, 22,486 transcripts with an N50 value of 959 bp and 19,458 predicted proteins. Approximately 67% of the *E. oleracea* transcripts showed a high confidence homology with *E. guineensis* proteome. RMBT was approximately 61%, and the mean read coverage was approximately 70x (Appendix A). 

Altogether, the different bioinformatic analyses discussed 217 here point out very clearly that TransABySS implemented with an additive multiple *k*-mer de novo assembly approach was the most efficient method to generate a high-quality transcriptome of the non-model plant *E. oleracea* for which RNA-Seq data were produced on the Ion Proton platform.

### 2.3. Functional Annotation and Biological Classification of Transcripts

#### 2.3.1. Gene Ontology and KEGG Annotation of *E. oleracea* Fruit Transcriptome

The *E. oleracea* transcriptome was functionally annotated with Blast2Go software (v.6.0), using the subset *Viridiplantae* of Uniprot database and an e-value of 1 × 10^−3^ (Figure 2c). From the 22,486 transcripts, 19,603 (87%) have shown a significant similarity with a protein of the UniProt dataset. Transcript annotations are available in Appendix A. Top hit species distribution reveals sequence homology with *Musa acuminata* (7431 hits), *Ananas comosus* (4469 hits), *Anthurium amnicola* (1780 hits), *V. vinifera* (500 hits) and *E. guineensis* (403 hits). The Gene Ontology (GO) as a standard system provides a classification of genes in several categories: Biological Process, Molecular Function, and Cellular Component. For a given organism, GO provides a prediction of the function of a gene and its product(s) [48]. Approximately 70% (15,811) of the *E. oleracea* transcripts were assigned to one GO term at least. The most critical GO categories were Biological Process (46,188 terms), followed by Cellular Components (27,046) and Molecular Functions (28,701).

*E. oleracea* transcriptome was annotated using Blast2GO and comparing to the Kyoto Encyclopedia of Genes and Genomes (KEGG) and Enzyme Codes database to have a global view of metabolic pathways [49]. A total of 6919 transcripts were assigned to 153 KEGG biochemical pathways and Enzyme Commission number (EC number) was identified for 1711 sequences (24.7% of the biochemical pathways assigned genes) (Appendix A). The most represented KEGG categories are “metabolism” (6561 transcripts), “organismal systems” (153 transcripts), “environmental information processing” (103 transcripts), “genetic information processing” (99 transcripts), and “drug resistance: antimicrobial” (3 transcripts) 241 (Appendix A). As expected, “metabolism” is a common category: it includes a global process such as photosynthesis, respiration, and biosynthesis and degradation of plant bioactive compounds [50]. The “metabolism” category is described in details in Appendix A. It is worth noting that “carbohydrate metabolism” (1404 transcripts), “amino acid metabolism” (783 transcripts) and “nucleotide metabolism” (683 transcripts) are the most important sets of genes. In plants, carbohydrates are the most abundant organic compounds while amino acids are not only building blocks for proteins synthesis but also precursors of important pathways involved in plant growth and adaptation to the environment [51]. Furthermore, the GO category “biosynthesis of other secondary metabolites” was investigated to identified genes related to the *E. oleracea* fruit pigmentation. Twenty-three subcategories were annotated, of which the top 10 was represented in Appendix A. The “phenylpropanoid biosynthesis” (99 transcripts) was the most important, followed by “flavonoids biosynthesis” (42 transcripts). The subcategories “caffeine metabolism” (12 transcripts) and “flavone and flavonol biosynthesis” (12 transcripts) were the smallest groups (Appendix A). The annotation of the transcriptome presented herein is in full agreement with the presence of several classes of phenylpropanoids in *E. oleracea* fruits including flavonoids, phenolic acids and anthocyanins [5].

#### 2.3.2. Comparison of GO Annotations between *E. oleracea* and Two Other Palm Fruit Transcriptomes

Figure 4 shows such a distribution of *E. oleracea*, *E. guineensis,* and *P. dactylifera* fruit transcriptomes into GO terms at level 2. Datasets of genes expressed in *E. guineensis* and *P. dactylifera* fruits were obtained by mapping published NGS read datasets during fruit development [52] and considering 1 TPM as a cut-off of expression. The distribution of GO terms at level 2 is very similar for the all three palm fruit transcriptomes (Figure 4). The genes classified as Biological Process were numerous in the categories “metabolic process” (GO:0008152), “single-organism process” (GO:0044699) and “cellular process” (GO:0009987). In the Cellular 266 Component group, the major GO terms were the categories “cell” (GO:0005623), “organelle” (GO:0043226) and “membrane” (GO:0016020). Finally, in the Molecular Function group, “catalytic activity” (GO:0003824), and “binding” (GO:0005488) were the most highly represented GO terms.

#### 2.3.3. Predicted *E. oleracea* Proteome Annotation by Orthology Inference

The OMA pipeline was used to increase the annotation accuracy of *E. oleracea* transcripts, predicting gene orthology and paralogy by mapping these predicted protein sequences to documented orthology and paralogy groups. Due to the high computation time to infer HOG and OMA groups [53,54], the predicted *E. oleracea* fruit proteome was compared to five complete proteomes: *P. dactylifera*, *E. guineensis*, *S. lypersicum*, *V. vinifera* and *A. thaliana*. *P. dactylifera*, *E. guineensis* are two palm trees with a complete and annotated genome (Figure 1, Table 1), but its fruits are different in morphology and metabolite composition. *S. lypersicum* was chosen as a model for fruit development [55] and *V. vinifera* as a model for anthocyanin biosynthesis and its regulation [56]. *A. thaliana* is a dicot, evolutionary distant from palm trees, but has the complete plant genome with well-curated gene annotations. *A. thaliana* is therefore very informative to infer gene function from data mining [57] and has been used to characterize structural genes involved in the anthocyanin biosynthesis pathway [58].

The results generated by the OMA pipeline were represented by taxonomic levels (Figure 5). The comparison between two levels revealed identical, duplicated, lost, novel or singleton genes. The number of proteins found in the six species was highly variable, from 41,887 in *E. guineensis* to 27,489 in *A. thaliana* for complete proteomes, and 19,475 for *E. oleracea* fruit proteome. In the latter case, the low number of sequences may be explained by the fact that the *E. oleracea* dataset was based only on protein sequence prediction from the fruit transcriptome and not from the whole genome as the other plants. The number of predicted proteins common to all six species annotated as mesangiospermae in Figure 5, is approximately 11,500. The protein number of palm trees, at the Arecaceae node, is higher than that found at the pentapetalae node, due to a higher level of duplication and not to novel genes (Figure 5). The Arecoideae node is con comparing to the Arecaceae node, suggesting the presence of 5381 new paralogs and the loss of the 1686 genes (Figure 5). A high proportion of duplicated, lost and singleton genes was found in the case of *E. oleracea*, with 2750, 11,453 and 4853 genes, respectively (Figure 5). 

The lost gene category in *E. oleracea* should include the lost and missing genes because the data were taken from the fruit transcriptome only. The duplicated gene category was probably a result of duplication events in palm trees [47]. Next, the distribution of Ks was plotted for paralogs of *E. guineensis*, *P. dactylifera,* and *E. oleracea*: two peaks appeared distinctly, indicating two successive events of duplication (Figure 6). The Ks distribution for a single-copy ortholog dataset based on *E. guineensis* and *E. oleracea* transcriptome showed that after the second duplication, a continuous divergence was observed, as otherwise reported in the case of mangrove palm by He et al. (2015) [47]. The singleton category contained genes not classified in any ortholog pair and restricted only to a lineage. A GO term enrichment test was performed with *E. oleracea* singletons compared to the whole-fruit transcriptome (Appendix A). In the biological process, three categories were related to primary shoot apical meristem, auxin transport, and callose synthesis. This result suggests that *E. oleracea* has specific and restricted genes related to developmental and stress-responsive mechanisms, such as plasmodesmatal trafficking and cellular division. Other categories of enrichment are related to plant stress response, as chaperone Hsp90. In parallel, detection of genes with a positive selection was performed using single-copy orthologs shared between *E. oleracea*, *P. dactylifera*, *A. thaliana* and *E. guineensis.* Three *E. oleracea* transcripts were detected with a positive selection (Appendix A) and were annotated as a vacuolar-sorting receptor, nucleobase-ascorbate transporter, and haloacid dehalogenase-like hydrolase. These proteins are involved in trans-Golgi network, transport and homeostasis of nucleotides, and Pi pool homeostasis. Positive selection should have occurred due to environmental adaptation: for example, the phosphate is immobile and unavailable for plant use and therefore is growth-limiting in the mangrove soil [61]. The OMA, Ks distribution and positive selection detection analysis suggest the existence of multigenic families with several paralogs and genes restricted only to *E. oleracea*. Further investigation, for example for specialized metabolism, should lead to understanding better the adaptation of this palm to tropical and waterlogged conditions. Transcriptomics approaches are often used to discover enzymes and transcription factors involved in specialized metabolism from non-model plants [11,62]. Phenylpropanoids are compounds derived from the amino acid L-phenylalanine and include diverse specialized metabolite pathways, including the flavonoids, coumarins, and lignans [63,64]. *E. oleracea* fruits have a notably high content of anthocyanins [65,66], and previous KEGG annotation has revealed 99 sequences for this pathway.

The accurate phenylpropanoid pathway annotation performed in this work using OMA resulted in the identification of *E. oleracea* orthologs in *A. thaliana, Vitis vinifera, Solanum lycopersicum, E. oleracea, P. dactylifera*, and *E. guineensis* proteomes (Figure 7 and Appendix A). Genes encoding enzymes of the anthocyanin pathway were also identified in açaí transcriptome, such as enzymes of the phenylpropanoid pathway: phenylalanine ammonia lyase (PAL), cinnamic acid 4-hydroxylase (C4H), and coumarate CoA ligase (4CL); enzymes involved in the early steps of the flavonoid pathway: chalcone synthase (CHS), chalcone isomerase (CHI), flavanone 3-hydroxylase (F3H) and flavanone 3′-hydroxylase (F’3H); and enzymes implied in the late steps of the anthocyanin pathway: dihydroflavonol reductase (DFR) and anthocyanidin synthase (ANS). Additionally, a putative UDP-glucose-flavonoid 3-O-glucosyltransferase (UGT) was also identified (Figure 7). In *A. thaliana,* the highest number of isoforms was observed for PAL and F’3H with four and seven isoforms, respectively, as previously reported [58,67].

In the açai transcriptome, all deduced enzymes exhibited direct orthology with *A. thaliana,* except in the case of C4H and F3H (Figure 7). Between *V. vinifera* and *S. lycopersicum,* the number of isoforms was highly variable: up to fifteen and eight PAL isoforms, respectively, were found, while ten CHS isoforms for *V. vinifera* and seven F3′H isoforms for *S. lycopersicum* and *A. thaliana* were identified. PAL isoforms are encoded by a multi-gene family with an estimated twenty putative PAL genes in *S. lycopersicum* [68] or fifteen putative PAL genes in *V. vinifera* [69]. In the predicted palm proteomes five putative genes coding to C4H and ten coding to CHS were found in the *P. dactylifera* and *E. guineensis*, respectively (Figure 7). As described previously, the occurrence of many events is likely to explain the high number of genes for several biosynthetic steps. In all palm species, the number of isoforms of 4CL, CHI, DFR, and ANS is conserved. However, the number of genes found in *E. oleracea* was lower than in other palm species, mostly because these genes were detected in the fruit transcriptome only (and most probably functionally expressed in the fruit). In fact, transcriptome analyses have been successful in the identification of genes involved in anthocyanin biosynthesis in plants [70,71,72,73,74,75,76]. Additionally, the expression of multiple isoforms of anthocyanin biosynthetic genes has been shown during fruit ripening [70,75,76]. As described by Darnet et al. (2011) [77], *E. oleracea* fruit has a high content of vitamin E due to an active tocopherol pathway. Along these lines, all five enzymes involved in the biosynthesis of tocopherols were identified in açai, based on clear, direct orthology with *A. thaliana* enzymes (Figure 8, Appendix A). Interestingly, many splicing isoforms of many enzymatic proteins are predicted when palm species are compared (Figure 8). For example, the number of isoforms for HGA phytyltransferase and 2-methyl-6-phytyl-1, 4-benzoquinol methyltransferase genes are the same between *P. dactylifera* and *E. guineensis*. This suggests that the genetic diversity observed in the tocopherol pathway results from splicing variations and not gene duplications. The same observation was made in the case of the early steps of the isoprenoid pathway, which displayed different splicing forms that were functionally active [78].

#### 2.3.4. Polymorphic Genic-SSR Detection and Transferability in Other Palm Trees

Using MISA tool, 904 EST-SSRs were found in 832 transcripts (Appendix A). The most abundant SSR types were trinucleotides (64.5%) and dinucleotides (33.6%) (Table 2). The most frequent repeats identified were AG/CT for dinucleotides, with 133 counts representing 44% of all dinucleotides. In the trinucleotide class, CTC/GAG repeats 366 were the most frequent (11%) (Appendix A). The SSR motif distribution is slightly different than reported by He et al. (2011), in other species of palm trees. For *E. oleracea* dinucleotide SSRs, the AG/CT and GA/TC are well represented with approximately 40%, and for other palm trees, the AG/CT represents approximately 82% (Appendix A) [47]. The trinucleotide SSR more abundant is CTC/GAG in *E. oleracea* transcriptome (12%) and CCG/CGG in other palms, approximately 25% (Appendix A) [47]. The enrichment in trinucleotide SSRs is related to the position in the genic sequence and is favored by the fact that triplets respect the reading-frame integrity [79]. For 620 SSRs positions, the PrimerPro pipeline has successfully predicted primer pairs for PCR amplification, thus providing molecular tools for breeding and diversity studies (Appendix A). CandiSSR pipeline implemented in this study has shown one trinucleotide EST-SSR polymorphism between *E. oleracea*, *E. guineensis* and *P. dactylifera,* eight trinucleotides between *E. oleracea* and *P. dactylifera*, and 18 between *E. oleracea* and *E. guineensis* (Table 2, Appendix A). The significant higher transferability rate from *E. oleracea* to *E. guineensis* than to *P. dactylifera* should be explained by the proximity of species in the phylogenetic position [8]. This new gene-based SSR dataset for *E. oleracea* opens new perspectives to analyze the genetic diversity in the *Euterpe* genus and, in other palm trees.

## 3. Discussion

The work described in this report is establishing the fruit of *Euterpe oleracea* as an attractive new model to study berry ripening and the concomitant accumulation of bioactive metabolites, proteins, and fibers. A primary objective of such a goal is the implementation of molecular resources and genetic information for further studies in fruit physiology and biochemistry and applications in plant breeding or agroecology. Here, this objective is achieved using an original and exhaustive de novo characterization of transcriptomes of *E. oleracea* using a 390 tool developed as AMAZOME. It is worth noting that the generation of such a tool represents a challenge for each new species of economic and agricultural interest, and amenable to genomics. The phenology of *E. oleracea* characterizes the fruits (or berries) as small in size, with a thin skin, black with a substantial UV-absorbing anthocyanin content, and covered by a thin wax layer at the late mature stage (Appendix A) [80,81]. At early stages, the fruit is rather small; the pulp is very thin, hard, and completely green due to the presence of chlorophylls and some polyphenols [65]. A critical step for developing a transcriptomic approach was first the optimization of the RNA extraction process to obtain a suitable quantity of high-quality RNA, for each stage. Differences in yield of total RNA have been measured depending on the ripening stage of fruits and the associated abundance of polyphenols, which can coprecipitate and degrade RNA samples [82,83]. The best total RNA quality was found by extracting the whole pulp of each fruit at each chosen ripening stage: more than 10 μg total RNA per fruit was obtained with a RIN superior to 7. Second, the sampling of fruits in native Amazon floodplain forest represented another challenge because of the structure of the palm inflorescence and also because of the small size of flowers that renders the time after pollination (in days after the emergence of female flowers) difficult to assess. Four maturation stages based on fruit size, color pattern and color intensity were defined for the samplings, as previously described [4]. Finally, the completeness and representativeness of fruit transcriptomes were ensured by sampling four individual inflorescences chosen from four independent palm trees, and this for two different varieties (Appendix A).

In this study, we took advantage of the Ion Proton sequencing platform that produces long reads and a strand-oriented system, which facilitates cDNA sequence reconstruction in the case of de novo sequencing [84]. In fact, a transcriptome assembly needs to be optimized on a case-by-case basis, for each given species and NGS platform [16]. A comparative assessment of de novo approaches carried out in plant transcriptomics indicated that assembler tools and parameters were diverse [18,19,85,86]. A reasonable explanation for this is undoubtedly linked 415 to the type of biological samples and also to the intrinsic characteristics of a genome, such as GC content or the occurrence of repeated sequences, and to the choice of technical options for cDNA libraries preparation and sequencing protocols. Indeed, the cDNA fragmentation and library amplification steps are critical and may affect sequencing efficiency. Each NGS system has its specific error spectrum that includes base substitution or size variation in homopolymer regions [87]. In our study, an assembler designed with specific parameters matching well Ion Proton technology was compared with more generic tools.

Interestingly, we found that the *k*-mer size was an important parameter that affected transcriptome reconstruction. The best results were obtained for the *k*-mer value of 39–43, similarly to studies reported recently by others using Ion Proton technology [88]. Other studies using Illumina sequencing and trinity tool assembly used a *k*-mer value of 25 [44,89]. The higher *k*-mer value defined in our study is probably due to the higher read size obtained with Ion Proton compared to Illumina platforms. A low *k*-mer value (20–23) is preferred for the assembly of poorly expressed transcripts, whereas a higher value (50 or more) helps well the accurate reconstruction of highly expressed transcripts [20]. In fact, the actual trend for de novo assembly is to merge several transcriptomes reconstructed at the different *k*-mer size, increasing then the completeness of a unique predicted transcriptome [19,21,90]. The combination of transcriptomes as a post-processing step offers the considerable advantage of joining datasets and removing sequence redundancy [15]. Specifically, tools such as STM, iAssembler, cd-hit, and Evidential Gene were developed for this purpose, albeit with a different rationale. For instance, cd-hit was developed to remove nucleotide sequence redundancy whereas other assemblers use a multistep analysis based on sequence comparison of nucleotides and of amino acids for the same purpose. The data obtained in this project also indicated that a transcriptome merged with Evidential Gene from multiple *k*-mer strategies exhibit superior metrics (such as the N50) than a transcriptome assembled at one *k*-mer value only. Post-processing steps reducing the total transcript number from 100,000 to 20,000–30,000 consolidated transcripts in agreement with the gene number predictions offers the considerable advantage to carry out a functional annotation of a non-model organism on very solid bioinformatic analysis.

The annotation and mining of de novo sequenced and assembled transcriptomes are critical to developing further functional plant genomics [27,91]. The typical approach is to annotate the dataset using a BLASTX comparison with plant protein sequence databases, such as the *Viridiplantae* section of UniProt [11,27]. This approach is however limited by partial or poorly annotated genomes and transcriptomes. In fact, highly curated and annotated genomes, such as those of *A. thaliana* or *Oryza sativa*, are the most informative to obtain sound functional inferences for non-model plant annotation, and this independently from the evolutionary distance between species [27,91]. The goal of our research program is to established genetic resources mainly to explore metabolic pathways associated with the deposition in particular organs of a large number of bioactive compounds. In plants, the rise of chemodiversity is due to gene duplication. Paralogs are responsible for modification and branching of metabolic pathways in specialized cell-types or organs, at given developmental stages, and under defined environmental conditions [35]. An accurate prediction of orthologs and paralogs between species is necessary to have a functional view of transcriptome and a genome. Among many ortholog prediction tools, OMA was chosen because it complied with two criteria: firstly, the high accuracy of the prediction, and secondly, the prediction of one-to-one ortholog between species [32,36]. In the case of multigenic gene families often implied in plant metabolic diversity, the identification (among many isoforms) of a direct ortholog already characterized in a species of reference and paralogs represent valuable pieces of information [92]. The OMA tool was applied to five predicted proteomes including *E. oleracea*, using an extended timeframe of two months of calculation to obtain series of pairwise orthology relationships (Appendix A) and Orthoxml files (Appendix A). Most importantly, in-house scripts were developed as user-friendly programs for the biologists to retrieved the functional annotation for each transcript (Appendix A). In the context of the richness of the Amazonian palm ‘açai’ in bioactive compounds, our work provides the identification of genes at play in the biosynthetic pathways for anthocyanin and tocopherol production in the fruit (Figure 7 and Figure 8). Furthermore, when applied to the model plants *A. thaliana* and *S. lycopersicum*, the bioinformatic pipeline AMAZOME implemented here predicted orthologs and paralogs precisely as reported [58,93]. In conclusion, *E. oleracea* transcriptome was fully annotated in work described here thanks to the design of a bioinformatic pipeline that we named AMAZOME that allowed a refined prediction of orthologs and paralogs for further functional genomics. This genetic resource will undoubtedly broaden the field of fruit ripening and physiology and represent a valuable tool to study carbohydrate partitioning and the production of bioactive compounds.

## 4. Methods and Materials

### 4.1. Plant Material and RNA Isolation

*Euterpe oleracea* bunches were collected from wild palms in the native floodplains of Amazon estuary (1.73972 S and 495 48.91697 W). A total of sixteen *E. oleracea* drupes were selected according to the four different ripening stages described by Rogez et al., 2011 [66] and originated from four different palm trees. Two palm trees are from black the açai variety, with high anthocyanin content and two from white variety, without anthocyanin content [3,94]. Immediately after collection, the exocarp and mesocarp of each drupe were manually removed and stored in RNA later solution (Ambion, Carlsbad, CA, USA) at −80 °C. Total RNA was extracted with rNeasy Plant Mini Kit (Qiagen, Hilden, Germany). The quality and quantity of total RNAs were checked using a Nanodrop Lite Spectrophotometer (Thermo Fisher Scientific, Waltham, MA, USA) and a TapeStation 2200 bioanalyzer (Agilent, Santa Clara, CA, USA). Only samples with RIN values higher than 7.0 were used for further analysis. 

### 4.2. Strand-Specific mRNA-Seq Sequencing

The mRNA-Seq library was constructed following Ion Total RNA-Seq Kit v2 specifications and using kits and reagents from Thermo Fisher Scientific (Waltham, MA, USA). Briefly, Poly (A)+ transcripts were isolated from 15 μg of total RNA using a Dynabeads mRNA Direct Micro Purification Kit. After this step, the samples were subjected to enzymatic fragmentation with rNase III for 1 min and 50 s at 37 °C in a thermo-block. The fragment purification was performed using a Magnetic Bead Clean-up Module as described in the Ion Total RNA-Seq v2 kit protocol. The cDNA preparation was made on the AB Library Builder System, respectively. The template preparation, the PI chip loading and sequencing step were automated using the Ion Chef system and Ion Proton sequencer, respectively. Libraries were sequenced using four PI chips (version 3), and all runs were merged and deposited in SRA-NCBI under SRR5330846 accession number. 

### 4.3. Bioinformatics Analysis

The pre-processing step was performed with Trimmomatic and cd-hit-EST tools [95,96]. Trimmomatic parameters were adjusted to Ion Proton reads and were the following: head crop: 7, crop: 200, leading: 3, trailing: 3, sliding window: 4:20 and minlen: 50. Cd-hit-EST was used with default parameter for removing identical reads from the dataset. The optimization of the transcriptome shotgun assembly (TSA) was based on three kinds of assembler. Firstly, assemblers are chosen with specific parameters for Ion Torrent/Proton technology, as MIRA 4.0 and rnaSPAdes 3.10 [97,98]. Secondly, the assemblers generic for all NGS platform, Velvet/Oases, Trinity 2.4, Trans-Trans-aBySS 1.5.5, SOAPdenovo-Trans 1.03 were used for the optimization [44,99,100,101]. Thirdly, a commercial software, namely, CLC Genomics Workbench 6.5, was used to compare results (http://www.clcbio.com/products/clc-main-workbench/), accessed on 12 February 2022. Multiple additive *k*-mer approach was based on a two-step protocol. First, assemblies were generated with a specific assembler using *k*-mer size from 19 to 99 and secondly, assembly results were concatenated and combined with the tr2aacds script of mRNA Transcript Assembly Software (Evidential Gene pipeline) to remove redundancy and extend transcripts in super-transcripts, as described by Nakasugi et al. (2014) [21]. Transcriptome characterization was based on metrics obtained with the Transrate tool [22] and BUSCO [23]. Additional in-house shell pipeline was used to evaluate the RMBT parameter. 

### 4.4. Functional Annotation

*E. oleracea* TSA functional annotation was performed using Blast2Go software and an initial BLASTX homology search with an e-value of 1 × 10^−3^ and the *Viridiplantae* section of the UniProt database [48,102]. Additionally, the protein signatures (InterProScan), EC numbers and KEGG annotations were assigned by Blast2Go software [48]. *E. oleracea* proteome was predicted using Transdecoder tool (https://transdecoder.github.io/), accessed on 12 February 2022. The orthology assignment between proteomes was performed using OMA [36]. Proteomes of *Arabidopsis thaliana, Solanum lypersicum, Vitis vinifera*, *E. guineensis* and *P. dactylifera* were retrieved from TAIR, under version 10.36, and NCBI, under accession GCF_000188115.3_SL2.50, GCF_000003745.3_12X, GCF_000442705.1_EG5, GCF_000413155.1_DPV01, respectively. In-house shell scripts and the FamilyAnalyzer tool (https://github.com/DessimozLab/familyanalyzer), accessed on 18 February 2022 were used to convert OMA results into annotation tables and to generate orthology statistics to identify identical, duplicate, novel and lost genes between taxa and ancestral nodes. For the different metabolic pathways, candidate genes were automatically retrieved with a shell script from OMA prediction and by comparison with the list of *A. thaliana* genes that were already functionally characterized. The detection of positive selection with site models was performed using the POTION pipeline based on CodeML and PAML tools [103]. The Ks distribution in transcriptome was determined using KaKsCalculator and ParaAT pipeline [104,105].

### 4.5. Polymorphic Genic-SSR Detection and Transferability Prediction

*E. oleracea* transcriptome was mined to identify polymorphic genic-SSR as described by Gordo et al. 2012 [41]. The SSR transferability was performed using CandiSSR pipeline with default parameters [106]. The transcriptome dataset for comparison was composed by GCF_000442705.1_EG5 and GCF_000413155.1_DPV01 NCBI access number, for *P. dactylifera* and *E. guineensis*, respectively.

## 5. Conclusions

*Euterpe oleracea* palm is endemic to the Amazon region, and its fruits possess nutritional and medicinal properties. Approximately 255 million single-end-oriented reads were generated on an Ion Proton NGS platform combining fruit cDNA libraries at four ripening stages. The de novo transcriptome assembly using the multiple *k*-mer approach showed that TransABySS assembler had best results, with an N50 of 959 bp, a read coverage mean of 70x, a BUSCO complete sequence recovery of 36% and an RBMT of 61%. The fruit transcriptome dataset included 22,486 transcripts representing 17.9 Mbp, of which 87% had significant homology with other plant sequences. The GO classification of transcripts from *E. oleracea* showed similar categories to that in *P. dactylifera* and *E. guineensis* fruit transcriptomes. The pipeline AMAZONE was developed whose was observed the annotation of putative genes from anthocyanin and tocopherol pathways. In the anthocyanin pathway was showed a high number of paralogs, such as in grape, whereas the tocopherol pathway showed conserved gene number and the prediction of several splicing forms. The annotated molecular dataset of *E. oleracea* constitutes a valuable tool for further studies in metabolism partitioning and opens new great perspectives to study fruit physiology with açai as a model in several areas such agriculture to develop açai varieties resistant to diseases and pests, in molecular biology to identify genes involved in biological processes, and in biotechnology to develop genetic improvement techniques for produce bioactive compounds on a large scale through genetic engineering.

## Figures and Tables

**Figure 1 ijms-24-09315-f001:**
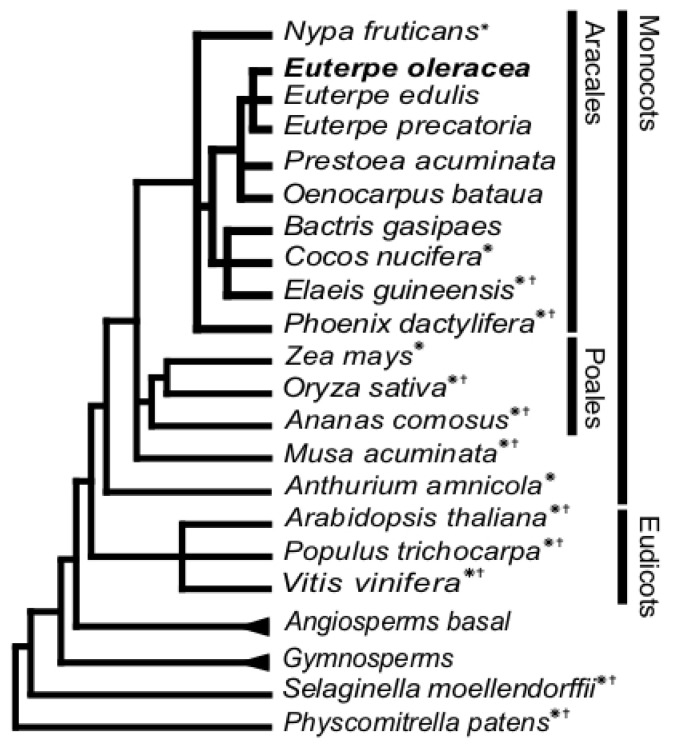
Phylogenetic position of *Euterpe oleracea*. *E. oleracea* and other palm trees from the Aracales are related to monocots and distantly related to eudicots. * Species with known transcriptome data; ^†^ species with known genome data.

**Figure 2 ijms-24-09315-f002:**
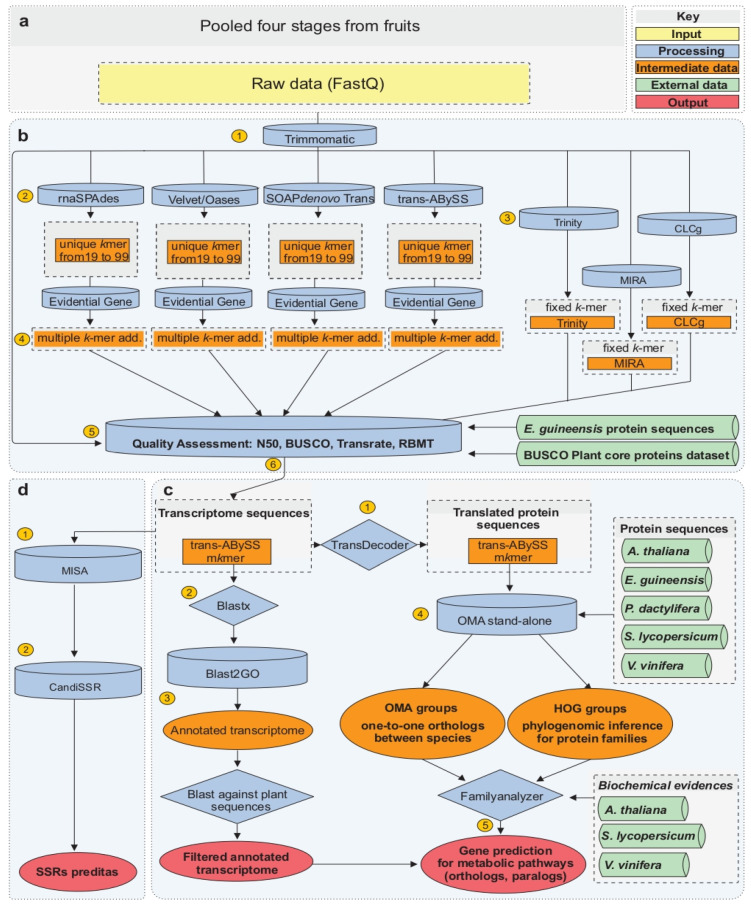
Overview of bioinformatic data processing. The pipeline developed for *E. oleracea* fruit transcriptome sequencing (**a**), de novo assembly ((**b**), steps 1–6), annotation ((**c**), steps 1–5) and data mining (**d**) is shown here and called AMAZONE. Key—yellow: input data; blue: processing steps; orange: intermediate data/files produced during the process; green: data from public databases; red: final output data.

**Figure 3 ijms-24-09315-f003:**
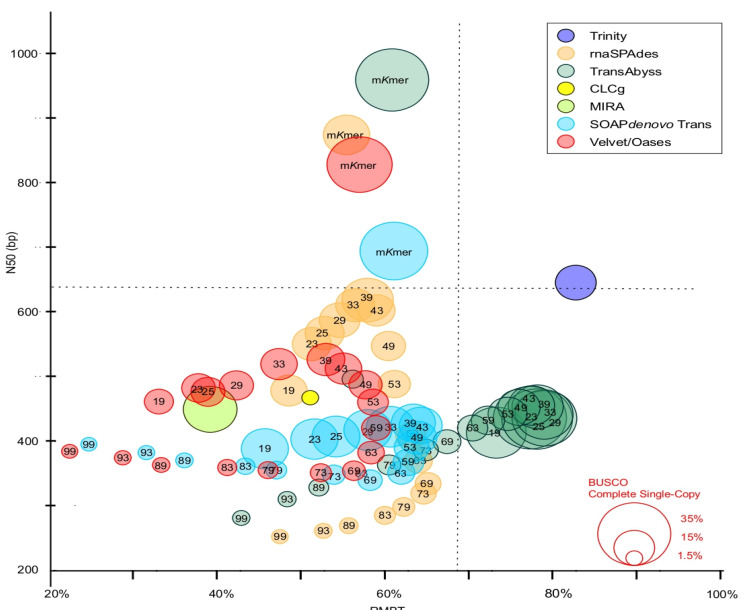
Completeness of açai transcriptomes obtained by de novo assembly optimization. *Y*-axis: N50 value; *X*-axis: reads mapped back to the transcripts; k: *k-mer* size; m*kmer*: multiple *k-mer* additive approach (*k-mer* from 19 to 99).

**Figure 4 ijms-24-09315-f004:**
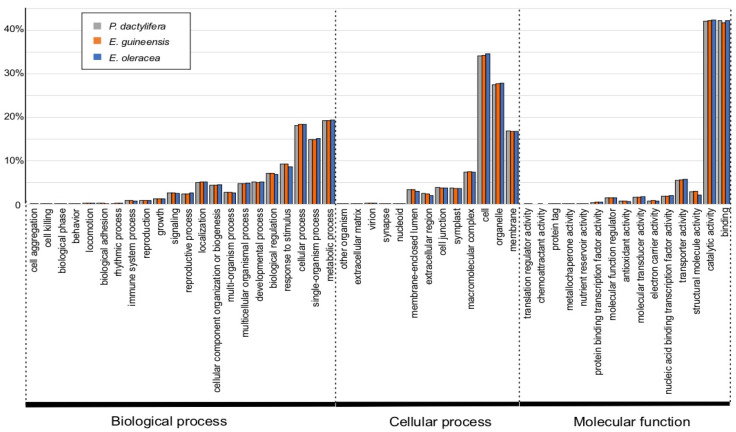
Gene Ontology statistics of *Euterpe oleracea* fruit transcriptome and transcripts expressed in *Phoenix dactylifera* and *Elaeis guineensis*. Gene Ontology classification of the fruit transcriptomes was based on BLASTX homology search in UniProt section *Viridiplantae*. The categories Biological Process, Cellular Component and Molecular Function, at the second level, were represented in the histogram.

**Figure 5 ijms-24-09315-f005:**
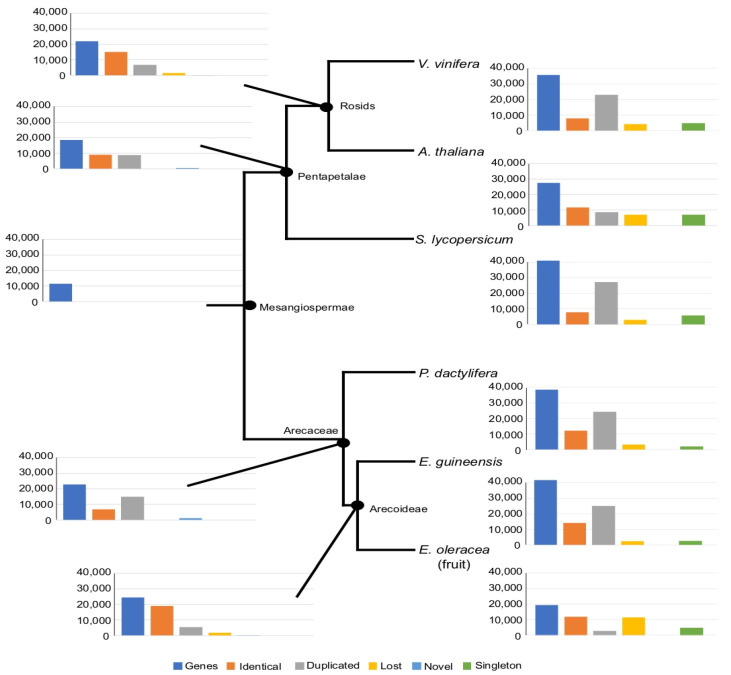
Evolutive orthogroup comparison among *Euterpe oleracea, Phoenix dactylifera, Elaeis guineensis*, *Solanum lycopersicum, Vitis vinifera,* and *Arabidopsis thaliana* proteomes. The ortholog inference among proteomes was performed 943 using the Orthologous Matrix (OMA) pipeline. The analysis of HOG groups, groups of proteins defined for specific taxonomic ranges, is used to identify genes directly descended from an ancestral (identical) and the novel, lost and duplicated genes. The singleton genes are restricted to a species and not classified in ortholog pair.

**Figure 6 ijms-24-09315-f006:**
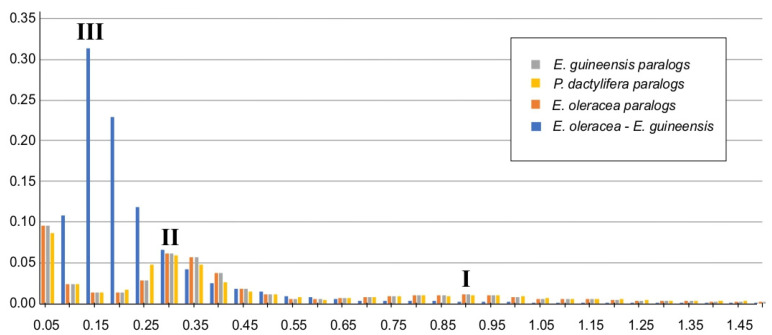
Whole-genome duplication events in *Euterpe oleracea*, *Phoenix dactylifera, Elaeis guineensis*. The distribution of synonymous substitution rate (Ks) of the paralogs is displayed in grey, yellow and orange for *Euterpe oleracea*, *Phoenix dactylifera* and *Elaeis guineensis*, respectively. Peaks I and II indicated two whole-genome duplication events in all three species. The blue bars showed the distribution of Ks of the orthologs between the *E. oleracea* and *E. guineensis*. Peak III is an indication of recent and continuous divergence between the two species after duplication (peak II).

**Figure 7 ijms-24-09315-f007:**
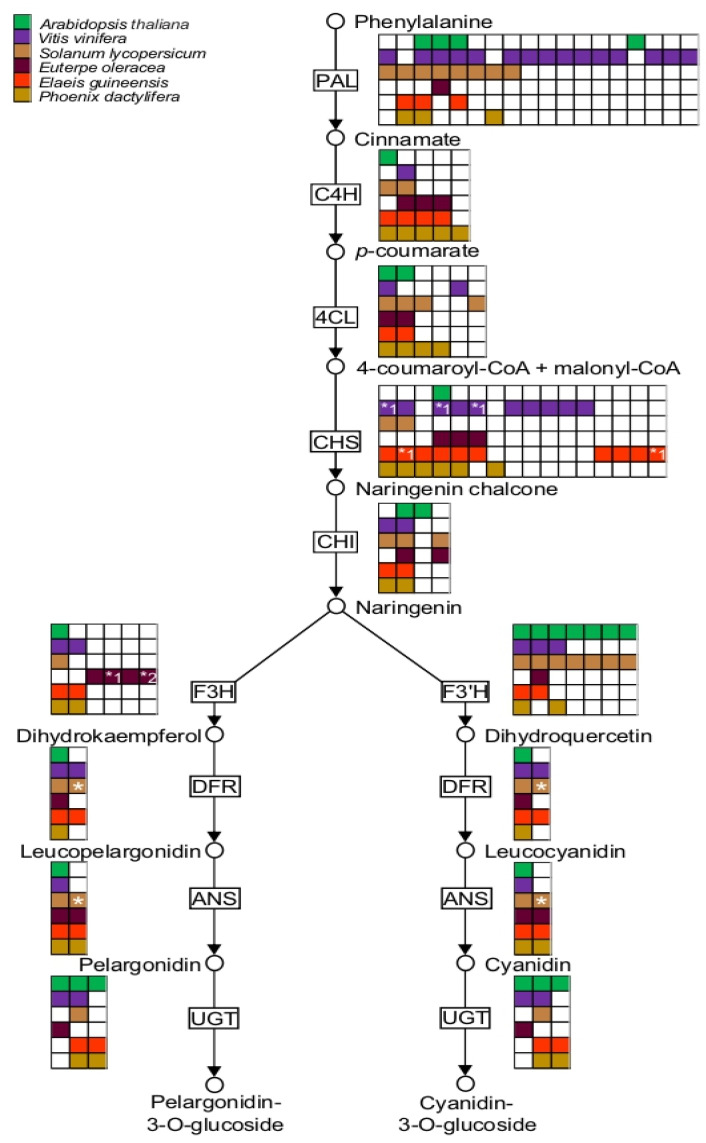
Simplified anthocyanin biosynthesis in Euterpe oleracea and orthogroups identified for each enzyme in *Arabidopsis thaliana*, *Vitis vinifera*, *Solanum lycopersicum*, *Euterpe oleracea*, *Phoenix dactylifera*, *Elaeis guineensis* and proteomes. ARATH: *Arabidopsis thaliana*; VITVI: *Vitis vinifera*; SOLLC: *Solanum lycopersicum*; EUOLF: *Euterpe oleracea* fruit; ELAGV: *Elaeis guineensis* PHODC: *Phoenix dactylifera*. Enzyme abbreviations: PAL, phenylalanine ammonia lyase; C4H, cinnamate 4-monooxygenase; 4CL, 4-coumarate CoA ligase; CHS, chalcone synthase; CHI, chalcone isomerase; F3H, flavonoid 3-hydroxylase; F3′H, flavonoid 3′-hydroxylase; DFR, dihydroflavonol 4-reductase; ANS, anthocyanidin synthase; UFGT, UDP Glc-flavonoid 3-O glucosyltransferase. * Different splicing isoforms.

**Figure 8 ijms-24-09315-f008:**
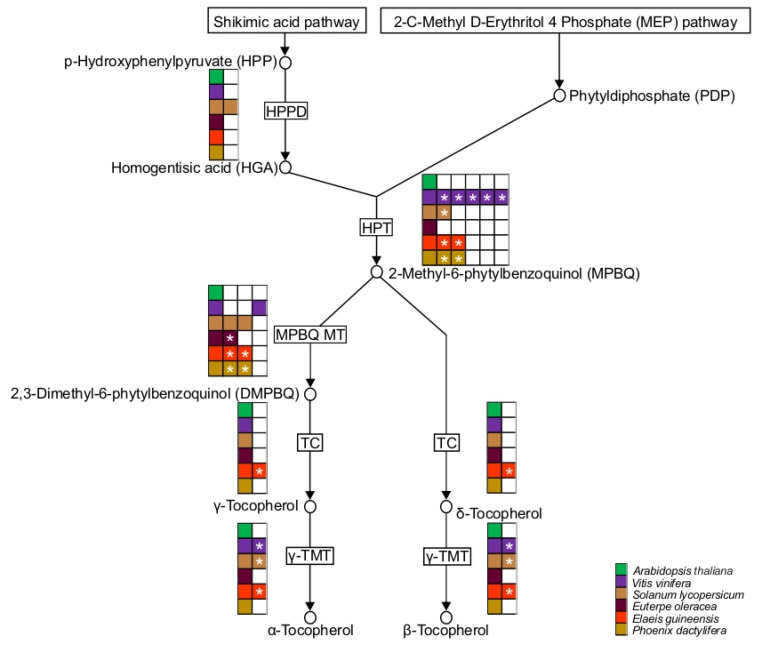
Simplified tocopherol biosynthesis in *E. oleracea* and protein orthogroup prediction for each enzyme in *Arabidopsis thaliana*, *Vitis vinifera*, *Solanum lycopersicum*, *Euterpe oleracea*, *Phoenix dactylifera* and *Elaeis guineensis*. ARATH: *Arabidopsis thaliana*; VITVI: *Vitis vinifera*; SOLLC: *Solanum lycopersicum*; EUOLF: *Euterpe oleracea* fruit; ELAGV: *Elaeis guineensis*; PHODC: *Phoenix dactylifera*. Enzyme abbreviations: HPPD, p-hydroxyphenyl pyruvate dioxygenase; HPT, HGA phytyltransferase; TC, tocopherol cyclase; MPBQ, 2-methyl-6-phytyl-1, 4-benzoquinol methyltransferase; γ-TMT, γ-tocopherol methyltransferase. * Different splicing isoforms.

**Table 1 ijms-24-09315-t001:** Transcriptome and proteome size of *Euterpe oleracea*, compared to *Phoenix dactylifera* and *Elaeis guineensis*. Data are from ^1^ Singh et al. (2013) [59]; ^2^ Al-Mssallem et al. (2013) [60]; ^3^ Torres et al. (2016) [9].

	*E. guineensis*	*P. dactylifera*	*E. oleracea*
Estimated Genome size	~1.8 Gb ^1^	~671 Mb ^2^	~4.2 Gb ^3^
Genome Reference NCBI	GCF_000442705.1_EG5	GCF_000413155.1_DPV01	n.a.
Genes	30,685	28,726	n.d.
Transcripts	41,801	38,432	n.d.
Predicted Proteome	41,887	38,570	n.d.
Predicted Fruit Proteome	17,778	18,139	22,486
% Annotated with GO	76%	75.8%	70.3%

**Table 2 ijms-24-09315-t002:** Polymorphic genic-SSR detection in *Euterpe oleracea* fruit transcriptome and transferability in other palm trees.

Total Number of EST Sequences Examined	22.517
Total number of SSR identified	904
Number of sequences contain SSR	832
Number of sequences contain more than 1 SSR	63
Dinucleotide	304
Trinucleotide	583
Tetranucleotide	9
Pentanucleotide	4
Hexanucleotide	4
Transferability EST-SSR	
Discrimination 3 species	1-tri
Discrimination *E. oleracea* and *P.dactylifera*	2-di; 6-tri
Discrimination *E. oleracea* and *E. guineensis*	1-di; 17-tri
Discrimination *E. guineensis* and *P. dactylifera*	1-tri

## Data Availability

Not applicable.

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
