# Peer review of "Elucidating the Mesocarp Drupe Transcriptome of Açai (Euterpe oleracea Mart.): An Amazonian Tree Palm Producer of Bioactive Compounds"

_ijms, 2023, doi:10.3390/ijms24119315_

Round 1

Reviewer 1 Report

1.      The annotation contains numbers that do not carry a semantic load: lines 15-18.

2.      The use of two specimens of the plant as a source of raw materials is considered insufficiently representative. Even for cloned plants, at least four specimens are usually taken.                                        

3.      Figure captions are too long. Perhaps the details should be moved to the main text.

Author Response

We thank the reviewer for the positive comments on this paper. Please see the attachment

Reviewer 2 Report

In the manuscript submitted for review, the Authors describe the mesocarp drupe transcriptome of Açai (Euterpe oleracea Mart.), an Amazonian tree palm. 

I find the topic of the manuscript extremely interesting, and "up to date", and the whole work is written well and thoughtfully. The reader's attention is undoubtedly drawn to carefully prepared figures.

My comments/questions:

1. For me, there was no clearly defined aim presented for the review of the experience.

2. While in the conclusion it would be important to know in which fields of science (medicine?) the results obtained by the Authors could be applied.

Author Response

(The authors gave the same response as above.)
